# Integrating human and ecological dimensions: The importance of stakeholders' perceptions and participation on the performance of fisheries co-management in Chile

**Milagros Franco-Meléndez**[1,2¤]*, **Jorge Tam**[3‡], **Ingrid van Putten**[4,5‡], **Luis A. Cubillos**[2‡]

**1** Programa de Doctorado en Ciencias con mención en Manejo de Recursos Acuáticos Renovables, Departamento de Oceanografía, Universidad de Concepción, Concepción, Chile, **2** Centro de Investigación Oceanográfica COPAS-Sur Austral, EPOMAR, Departamento de Oceanografía, Universidad de Concepción, Concepción, Chile, **3** Instituto del Mar del Perú, Callao, Lima, Peru, **4** CSIRO Oceans and Atmosphere, Castray Esplanade, Hobart, Tasmania, Australia, **5** Centre for Marine Socioecology, University of Tasmania, Hobart, Tasmania, Australia

¤ Current address: Departamento de Oceanografía, Universidad de Concepción, Concepción, Chile
‡ These authors also contributed equally to this work.
* milagrosfranco@udec.cl

## Abstract

Increasing attention is paid to the interdependence between the ecological and human dimensions to improve the management of natural resources. Understanding how artisanal fishers see and use the common-pool resources in a co-management system may hold the clue to establishing effective coastal fisheries policies or strengthening existing ones. A more comprehensive planning of the system will also have a bearing on how to reduce conflicts and strengthen social networks. We surveyed artisanal fishers and decision-makers to determine their perceptions about the Management and Exploitation Areas of Benthic Resources (known as MEABR) in Chile's Biobio region. We performed a field study from November 2018 to August 2019, applying a set of questionnaires to determine the ecological and human attributes that contribute to MEABR outcomes, and then constructed composite scores for those attributes according to a multidimensional scaling technique ("Rapfish"). We find that fishers have different perspectives: surprisingly, women highlighted that the institutional dimension was the most influential on MEABR performance, whereas men highlighted the ecological and economic outcomes. The decision-makers' role in the MEABR system was considered adequate, but communication and socialization of regulations were irregular. Results also showed that fishers expressed dissatisfaction with illegal fishing practices (poaching), productivity, profits, and conflicts inside and outside the MEABRs. Our study allowed us to better understand how the MEABR has developed in the region. We recommend strengthening local management strategies with particular attention paid to networking among stakeholders, including gender inclusive relationships.

**Data Availability Statement:** All relevant data are within the paper and its Supporting Information files.

**Funding:** MFM received partial funding from COPAS Sur-Austral (ANID PIA APOYO CCTE AFB170006) and Postgraduate scholarship from the Universidad de Concepción and the total funded support of EPOMAR (Laboratory for Evaluation of Marine Populations). The funders had no role in study design, data collection and analysis, decision to publish, or preparation of the manuscript.

**Competing interests:** The authors declare that they have no known competing financial interests or personal relationships that could have appeared to influence the work reported in this paper.

## Introduction

The purpose of fisheries management is to preserve and ensure sustainable use of the resource [1]. However it is still necessary to improve the performance of fisheries where there are serious problems of overfishing and the management system is deficient [2]. For this, several authors have suggested that integral monitoring should be improved for all fisheries, including artisanal fisheries [3, 4]. As fisheries are complex socio-ecological systems (SES), the interaction between the ecological components and the social realm that make up these systems needs to be strengthened [5, 6].

The need to improve fishery monitoring is particularly urgent in artisanal fisheries, as they produce most of the global landings for direct human consumption [7]. Artisanal fisheries are often recognized as an ancient form of social organization, production and trade settled for centuries along the coasts [8]. Nevertheless, data availability for artisanal fisheries is poor and it is therefore difficult to understand and effectively manage such fisheries, which are also poorly regulated [4]. In addition, the drive for short-term gains generates fishing impacts such as resource depletion and loss of biodiversity [9].

Consequently, the current dilemma facing fishery management is not just the limited understanding of the fish stocks or external pressures, but also knowledge of the human dimension and how it fits into fishery management [10–13]. There is a broad theoretical background for the role of the human dimension in fishery co-management [10, 14–19] but it is necessary to test this empirically with real fishing communities in order to understand and improve sustainable fisheries [20–22]. The Code of Conduct for Responsible Fisheries (CCRF, article 12) underlines the need to investigate and plan fishery measures, including social, legal and institutional aspects [23]. In other words, it is not possible to manage a fishery without managing the fishers, as the activity faces a diversity of problems which require holistic strategies to guarantee its sustainability [24–28].

Given this reality, various studies have shown that one of the strategies for improving management and sustainability of coastal fisheries is the Territorial Use Rights for Fisheries or TURF system [29–35]. The TURFs, also popularly known in Chile as the Management and Exploitation Areas for Benthic Resources-MEABR system (or AMERB, in its Spanish acronym), are designed to assign exclusive rights to artisanal fisher organizations in specific areas to harvest sustainably benthic species [36–38]. Fishers must provide baseline information and a management plan to target a single or groups of benthic species. In addition, private consultants hired by each fisher organization assess the standing stock biomass of the target species through field surveys to set an annual harvest quota authorized by SUBPESCA. The objectives of MEABR are a governance transformation through co-management and to enhance the sustainability of small-scale fisheries [39]. Thus, the TURF system, is a better administrative mechanism for controlling open-access resources, thus achieving an equitable distribution of benefits and maintaining social cohesion [29, 40–42]. Unfortunately, some TURFs have never produced the outcomes expected for various reasons due to scarce administrative, human and financial resources for surveillance of illegal fishing, among others [43, 44]. In this context, the main problem affecting the TURF system in Chile is the lack of a transdisciplinary approach that merges the ecological and social systems [45].

The MEABRs gained traction by the end-1990 and have had a considerable territorial impact over the years. In sites where MEABRs have been well applied, the density and size of target species increased substantially relative to open-access areas [46–53], generating positive outcomes for environmental sustainability [52, 54, 55] and socio-economic [42, 50, 55–60]. However, their management is still bedeviled by inefficiencies, because once a right is established, it is difficult to reverse or change it, even if unwanted outcomes occur [59, 61, 62]. For

example, some MEARBs managing surf clams [63, 64] could not solve the problems caused by the high temporal variability of this resource. As a result, the governance was abandoned and did not recover even when the resource itself recovered [59, 61, 65–70]. Nevertheless, one of the favorable effects of the MEABRs has been the empowerment of fishers' organizations, which have promoted collective actions among users [65, 71]. Collective action is broadly understood as "people joining together purposely for a common cause" [72]. Also, it could be confrontational, depending on the relationship among stakeholders [73]. Since their inception, MEABRs have been adapted to new regulations and fishing practices [74].

The Biobio region in south-central Chile, is an example of how most of the outcomes envisaged for the MEABR system have not been met [75]. In many cases, this lack of positive outcomes has in many cases caused frustration and the abandonment of MEABRs administered by artisanal fishers' organizations [76–78]. Management of the MEABRs is complex and fishers' responses can have unforeseen consequences leading to failure. As stated earlier, many examples of the MEABR system are being applied with no consideration of the socio-ecological systems upon which they were implemented [59, 61, 65, 79]. The lessons learned with the MEABR system in Chile could be helpful for the TURF system worldwide. For example, finding out why the regulations have not been followed or respected could help to avoid bad outcomes the system has generated so far. And could explain why the lack governance that prioritizes participatory management has contributed to the overexploitation and poor performance of small-scale marine fisheries [80].

Under these circumstances, understanding how fishers perceive and use the resources will have important repercussions for the management and local policies of the MEABRs. This will also have a bearing on how to reduce conflicts and strengthen social networks within the context of a more comprehensive planning of the system. One recent work [81] concluded that if fishers' organizations and consultants take as little part in the collective decision-making process as they have so far, it is unlikely that the necessary adjustments will be made to the system. Indeed, the different viewpoints of more men and women regarding management should be taken into account, in order to obtain a better understanding of the social interaction of women in an activity traditionally considered to be male-dominated [67, 82–84].

In Latin America, women play a fundamental role in fishing-related activities [85, 86]. Despite this, women are not an entirely homogeneous category of people in the sector, rarely found in statistics or documented in socioeconomic evaluations [87, 88]. Therefore women are subject to various degrees of inequalities in access to and control over productive resources, services, employment opportunities, and empowered participation in decision-making [89, 90]. They have been made invisible for many years. However, in the last few decades, the relevant role of women in contributing to new forms of sustainability through their work and traditional knowledge has been emphasized [83]. They gain more and more momentum and recognition in different instances and even as good managers and negotiators in the value chain and fishing system [67, 91–93]. Thus, in this changing scenario, it becomes relevant that gender mainstreaming should be an integral part of all small-scale fisheries development strategies [94]. Moreover, that leads to implementing new regulations on gender equity, such as modification of the current Law No. 18.892, General Law on Fisheries and Aquaculture in Chile (to be enacted), which seeks to incorporate gender quotas and opportunities to reach spaces for dialogue regarding fisheries management. That is why women's perceptions should include in any study of fisheries co-management performance.

Indeed, the perceptions of stakeholders (including fishers) are important indicators of the performance of the MEABR system, particularly when governance is deficient [95, 96]. Thus, for many of the world's fisheries, including the Chilean exclusive access regime, a more holistic evaluation of the ecological and human dimensions is necessary [27, 74, 97–101]. However, as

far as we know, few studies in the region have systematically documented the perceptions of fishers in a socio-ecological context. Thus "each coastal territory and the fishers who live in it are heterogeneous, consequently although changes may be seen as transverse, no generalizations can be made regarding the local responses" ([83] page. 177).

With the above statement in mind, we hypothesize that an individual's knowledge, social characteristics and perceptions about the MEABR system, influence their local fisheries' management expectations. This study proposes the following objectives: (i) to describe the participation of the various stakeholders (fishers and non-fishers) and their perceptions about local management of the MEABR system in Biobio region; (ii) to identify the problems and expectations of fishers concerning the MEABR system from the perspectives of gender, organizational role and accessibility (distance between the fishing cove 'caleta' and the MEABR); and (iii) to determine whether there are any unmet information needs, based on a more integrated evaluation of the different stakeholders. The text has been divided into three parts in which the proposed goals are addressed, before we finally make some recommendations regarding conservation and improving the MEABR system in the region.

## Materials and methods

### Study area

The Biobio Region (72˚30'0" W; 37˚0'0" S) covers 37,068.70 km$^2$ and represents 4.9% of Chilean territory. According to the 2017 National Census, the population is just over 2 million, with slightly more females (52%) than males. The Biobio Region has 15 coastal townships with four locally important fishing ports: Coronel, Lirquen, San Vicente, and Talcahuano. The landings in those ports amount to approximately 50% of total national landings. The Biobio Region has close to 20,000 registered artisanal fishers (74% men), and around 3,000 fishing vessels, mainly small boats (<12 m length) and decked boats (85%) [102]. Geographically, the coast of Biobio Region consists of the Gulf of Arauco, Concepcion Bay and Coliumo Bay, and three inhabited islands: Quiriquina, Santa Maria, and Mocha (Fig 1). Artisanal fishers are settled in both urban (54.3%) and rural (45.7%) localities [103], allowing them to develop their activities linked to fishing, seaweed collection and alternative activities such as commerce, tourism, agriculture and livestock among others.

The MEABRs located in Biobio Region cover approximately 26,000 hectares of ocean. The average size of an MEABR is 347 hectares, but varies significantly, with some reaching up to 4,096 hectares, such as the "Weste Isla Mocha" in Mocha Island and some less than one hectare, such as the "Rari" MEABR in Coliumo Bay. These differences in sizes (extent) of the MEABRs are relevant, as there is a large variation among the number of hectares per user, suggesting a wide variability in the physical characteristics of the areas assigned to each organization [104]. In addition, accessibility (e.g. those MEABR in isolated location or with easier access) has had implications on MEABR performance, as those MEABRs near urban centers have received more funding from governmental institutions [105], but are also more vulnerable to illegal fishing [4, 77, 106]. Other factors affecting MEABR performance are economic profitability and the biological productivity of the assigned MEABRs [104].These conditions influence fishers' relationship to resources, organization, functioning, and MEABR development.

The number of MEABRs in Biobio Region reached 127 by 2017, but only 78 were approved and awarded (Table 1), while 31 MEABRs were rejected (i.e. not approved nor awarded), and 18 were approved but not awarded [107]. The MEABRs are administered by 47 artisanal fishers' organizations, including unions, trade associations, indigenous associations, and

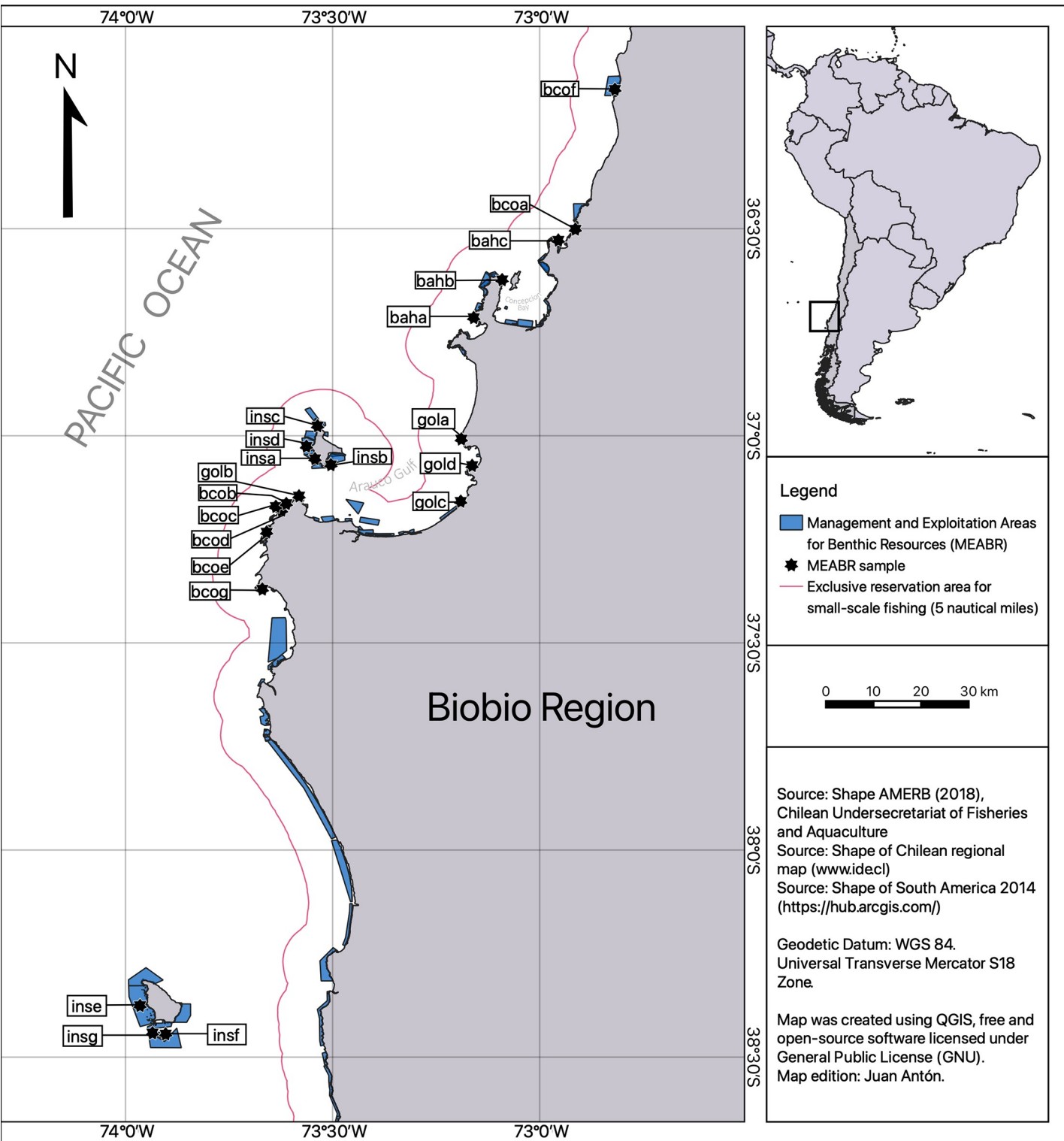

**Fig 1. Map of the study area: Fishing coves and MEABRs in the Biobio Region, Chile.** MEABR's Codes description (sampled): baha = San Vicente, bahb = Candelaria-Cantera, bahc = Rari, bcoa = Dichato, bcob = Punta Raimenco, bcoc = Bajo Rumena, bcod = Rumena, bcoe = Los Piures, bcof = Cobquecura sector A, bcog = Puerto Yana, gola = Maule, golb = Punta Lavapie, golc = Pueblo Hundido, gold = Laraquete, insa = Los Partidos, insb = Puerto Sur, insc = Pueblo Norte sector B, insd = Punta Cadena, inse = Weste Isla Mocha, insf = Quechol, insg = Quechol Sur. "Reprinted from Shape AMERB (2018), Chilean Undersecretariat for Fisheries and Aquaculture under a CC BY license, with permission from PLOS journal, original copyright 2021".

**Table 1. Status of the Management and Exploitation Areas of Benthic Resources (MEABR).**

| MEABR Status | MEABR Biobio Region (No.) | MEABR rest of Regions (No.) | MEABR Total Chile (No.) |
|---|---|---|---|
| Approved and awarded | 78 | 725 | 803 |
| Approved but no awarded | 18 | 272 | 290 |
| Rejected | 31 | 196 | 227 |
| Total | 127 | 1193 | 1320 |

Updated to 2017. Source: SUBPESCA (http://www.subpesca.cl/portal/619/w3-article-79986.html).

Approved and awarded: those MEABRs requested by a fisher's organization with the baseline study (or ESBA, in its Spanish acronym, which describes the initial status of MEABR as for the conservation status of species, ecosystem, socio-economic condition, bathymetry, among other aspects); Approved but not awarded: those MEABRs that are available to be used but to date no organization is interested, has expired or has been returned by a fisher's organization, or is pending a decision by competition among two or more organizations, Rejected: those MEABRs that did not get approval to be used because the process stopped earlier [109].

cooperatives. Men run nineteen organizations, men and women run 27, and one is exclusively run by women [108].

## MEABRs selection

From the 78 MEABRs approved and awarded, we selected 21 MEABRs according to the selection, stratification and sampling design described in the (S1 Text). These MEABRs were legally constituted and administered by artisanal fisher organizations (AFOs), located in either a gulf, a bay, along the coastline, or that were an island (see S1 and S2 Tables for additional information). The fisheries statistical data about the selected MEABRs was collected over 18 years (between 2000 and 2017). Finally, the sample size of each stratum in number of fishers is shown in Table 2.

## Surveys design

We considered two groups of stakeholders ("actors") in our analysis: fishers (comprised of leaders and members) and non-fishers, a group which includes decision-makers, academics and consultants. For fishers, the sample size determined according to the assigned organizations (Table 2) and its sample design is described in the (see S1 Text).

While contacting survey respondents, it became evident that it was impractical to interview fishers on the Hualpen peninsula because of the lack of voluntary participation, and another management area was chosen instead. This reluctance to participate may have been due to fears of revealing information or being misrepresented.

We interviewed fishers during November 2018 and March 2019, and a group of five interviewers conducted all the surveys. The interviews were face-to-face in the location of the participants' choice and we obtained prior consent from the interviewees (CEBB 747–2020). There were slight differences between the questionnaires for leaders and members (discussed in more detail below). We successfully surveyed five members and two leaders for 18 different fishers' organizations. Members included divers, ship-owners, and/or seaweed collectors. For two fisher organizations, only five respondents agreed to be surveyed, and in one organization, only one fisher agreed to participate. In total, 117 fishers took part in the survey, which is well above the required minimum sample of 88.

**Table 2. Classification and sample sizes of selected MEABRs by stratum.**

| Stratum | MEABR (No.) | AFO (No.) | Yield tons/m$^2$ (*1000) | Suitable substrate m$^2$/ 1000 | Area per member m$^2$/mbr (*10000) | Number of member (N$_h$) | Sample size by stratum (n$_h$) |
|---|---|---|---|---|---|---|---|
| 1 | 7 | 6 | 4.67–23.29 | 252.8–958.9 | 0.22–2.55 | 467 | 38 |
| 2 | 8 | 8 | 0.73–4.95 | 136.9–480.8 | 0.39–1.33 | 288 | 24 |
| 3 | 6 | 4 | 1.25–5.91 | 67.3–145.5 | 0.08–0.30 | 315 | 26 |
| Total | 21 | 18 | 0.73–23.29 | 67.3–958.9 | 0.08–2.55 | 1070 | 88 |

AFO, artisanal fisher organizations; mbr, member.

We also surveyed decision-makers because this will show us a wider picture of the situation regarding MEABRs at both regional and national levels. The sample selection was not random, but 'key informants' were selected. The intentional or convenience sampling [110] was carried out by the first author. This sampling ensured that participants with well-known expertise were chosen. For this survey, 20 expert participants were selected from the academy, the Undersecretariat for Fisheries and Aquaculture (SUBPESCA), National Fisheries and Aquaculture Service (SERNAPESCA), Fisheries Development Institute (IFOP), and specialist consultants' staff. Semi-structured interviews were conducted with the representatives of each institution. In addition, the questionnaire was applied to collect data about their perceptions and their experiences. The interviews were carried out between July and September 2019 and took approximately 1 hour.

## Questionnaires

For fishers (leaders and members) and decision-makers, the questionnaire included multiple-choice, scoring (on a 10-point Likert scale), and open-ended questions. The classification of attributes in six transdisciplinary dimensions from the Rapfish methodology [111] was adapted to assess the MEABR's performance in Biobio Region. The Rapfish method simultaneously appraises fisheries' status under six dimensions: ecological, technological, economic, ethical, social, and institutional. We included these dimensions in the study, each one supported by 7 to 10 attributes, totaling 51 scored attributes (see S3 Table).

The fisher survey applied to leaders consisted of two parts: 1) a description of the fisher organization (i.e. infrastructure) and relevant problems, benefits, and expectations for the MEABR; and 2) multiple-choice and rating questions concerning the Rapfish ecological and human attributes (see S1 and S2 Appendices for the questionnaires). The main difference between the member and leader surveys were questions related to family cooperation in fishing activities, education level, and participation in the fishing organization. The survey for decision-makers gathered some personal information (i.e. role, experience, etc.) then the ranking questions (see S3 Appendix). Each human and ecological attribute of sustainability was evaluated according to their degree of importance (less important-very important) and condition (unfavorable-very favorable) by allocating scores from zero to ten [111–113].

## Ethics statement

This study was approved by the "Ethical, Bioethical and Biosecurity Committee" of the Vicerrectoria de Investigación y Desarrollo, Universidad de Concepción (permit number CEBB 747–2020) and followed guidelines established by their ethics committee, which complies with national and international standards. The surveys included a written informed consent approved by interviewees, which acknowledged the research objectives and established that the survey was anonymous and that interviewees were free to choose to not answer questions.

## Data analysis

Questionnaire-based data obtained from decision-makers were ordinal data measuring different levels, from very positive to very negative. We explored the scorings from decision-makers and fisher's perceptions separately and compared them. We applied the Likert scale to measure respondents' attitudes to a particular question. We used the Likert package [114] in R software. In order to analyze differences between important factors which are fundamental for the collective action, surveillance and logistical aspects in MEABRs, we classified the survey respondents into three main factors: i) gender (male or female) to assess the participation of women within the organization [67, 72, 83, 85], ii) fishers' organizational roles (leader or member), their participation into organizations of which they are part [115, 116], and iii) MEABR accessibility or distance from fishing coves 'caletas' to the MEABR (accessible, less than 0.5 km or less accessible, more than 0.5 km), suggesting that human disturbances of many species and ecosystems are stronger in areas with higher physical accessibility [117–119]. We tested the differences between categories of factors using chi-square tests, one-way ANOVA analyses, and independent t-tests (depending on the data type). We used non-parametric tests for multiple independent samples that were not normally distributed and we analyzed the data with the R software [120] using the PMCMR library [121].

Besides looking at the difference in average scores for the Rapfish dimensions for fishers and managers, we also tested whether a range of other variables influenced the scoring of Rapfish dimensions. These variables were loosely grouped into three aspects: physical (i.e. distance from fishing, target specie), human (i.e. gender, age, marital status), and organizational (i.e. other activities, management type) (see S4 Table).

We performed a leverage analysis to determine any attributes sensitive to each dimension of sustainability employed in this study. For each of the attributes, we calculated the sum of squares of differences of the x -and y-scores. By doing so, we provide a standard error (S.E) expressing the leverage of each attribute. The S.E range in values is between 2% and 6%, according to Kavanagh and Pitcher [122]. We obtained the results of this analysis from the Rapfish version 3.1 software (www.rapfish.org).

## Results

### General description of fisher and decision-maker respondents

Of the 117 surveyed artisanal fishers, 77% were men (n = 90), 89% were less than 40 years old, 71% were younger than 20 when they started fishing, and 80% had more than 10 years of fishing experience. More than 80% of respondents did not complete primary and/or secondary education. Organizationally, 83% of respondents were members of a fishery organization (including divers, ship owners, diver assistants, and collectors). Nearly 80% of the respondents are dependent on fishing for their incomes (Table 3).

Of the 20 surveyed decision-makers, 80% were male, seven were resource managers (either from government or fisheries research institutions), four were private consultants, and nine were experts (academics).

### Perception about the socio-ecological system

Fishers and decision-makers were scored using transdisciplinary Rapfish ecological, technological, economic, ethical, social, and institutional dimensions (see S3 Table). Each of the Rapfish dimensions consists of several attributes. In this first part of the analysis, we use the average scores for each dimension, the role and influence of the individual attributes to the scoring (within each dimension).

**Table 3. Summary of demographic characteristics of fishers' respondents in Biobio Region.**

| Variables | Mean (SD) | Range |
|---|---|---|
| **Age (years)** | 51 (0.934) | 22–72 |
| **Experience (years)** | 19 (0.629) | 4–32 |
| **Number of household members** | 4 (0.158) | 1–11 |
| **Number of dependents in the household** | 2 (0.155) | 0–8 |
| Fishing activity ratio* | 0.77 (0.024) | 0.2–1 |
| **Variables** | Number | % |
| **Level of education** | | |
| Not attended school | 1 | 0.9 |
| **Not completed primary** | 45 | 38.5 |
| **Completed primary** | 37 | 31.6 |
| **Not completed secondary** | 16 | 13.7 |
| **Completed secondary** | 15 | 12.8 |
| **Higher education** | 3 | 2.6 |
| **Gender** | | |
| **Females** | 27 | 23 |
| **Males** | 90 | 77 |

*The proportion of income obtained from fishing activity in relation to other income earning activities, grouped by low, medium and high (more than 0.5 or 50%).

The decision-makers attributed significantly lower scores than fishers in all dimensions (One-way ANOVA, p-value<0.01), except for the institutional dimension (F = 4.153, p-value>0.01) (Fig 2). In other words, decision-makers rated the quality of the outcomes of MEABRs for each of these dimensions lower (or worse). Only fishers rated one dimension (technological) on average as good (a score of 7 or more), all other dimensions had an average score that was less than 7 for both groups. The managers attributed the lowest average scores to the ecological and ethical components. The ethical dimension also received the lowest average score by fishers.

## Perceptions on gender, organizational role, and accessibility

There was more difference in the average rating of ecological and social dimensions (ANOVA tests, p-value<0.01 and p-value<0.1, respectively) than the technological dimension (p-value>0.05). For instance, the score for the ecological dimension was significantly influenced by physical aspects such as the distance to where the MEABR was located (p-value<0.001), how accessible the MEABR was to the fishers (p-value<0.001), and the abundance of the main targeted species (p-value<0.001). Scoring for the social dimension was also significantly influenced by all physical variables (except for the location distance). This suggests that the physical characteristics of the MEABR influence the way respondents perceived the relative ecological and social outcomes of the MEABR.

Several human factors also significantly influenced the relative scoring of the transdisciplinary Rapfish dimensions. Gender influenced the scoring of the ecological and institutional dimensions with females attributing lower scores to the ecological and higher to the institutional dimension. It is worth mentioning that the sample size for females was small, as most of the fisher organizations surveyed consist mainly of men who catch Chilean abalone. Most females work on either seaweed collection or in other alternative activities.

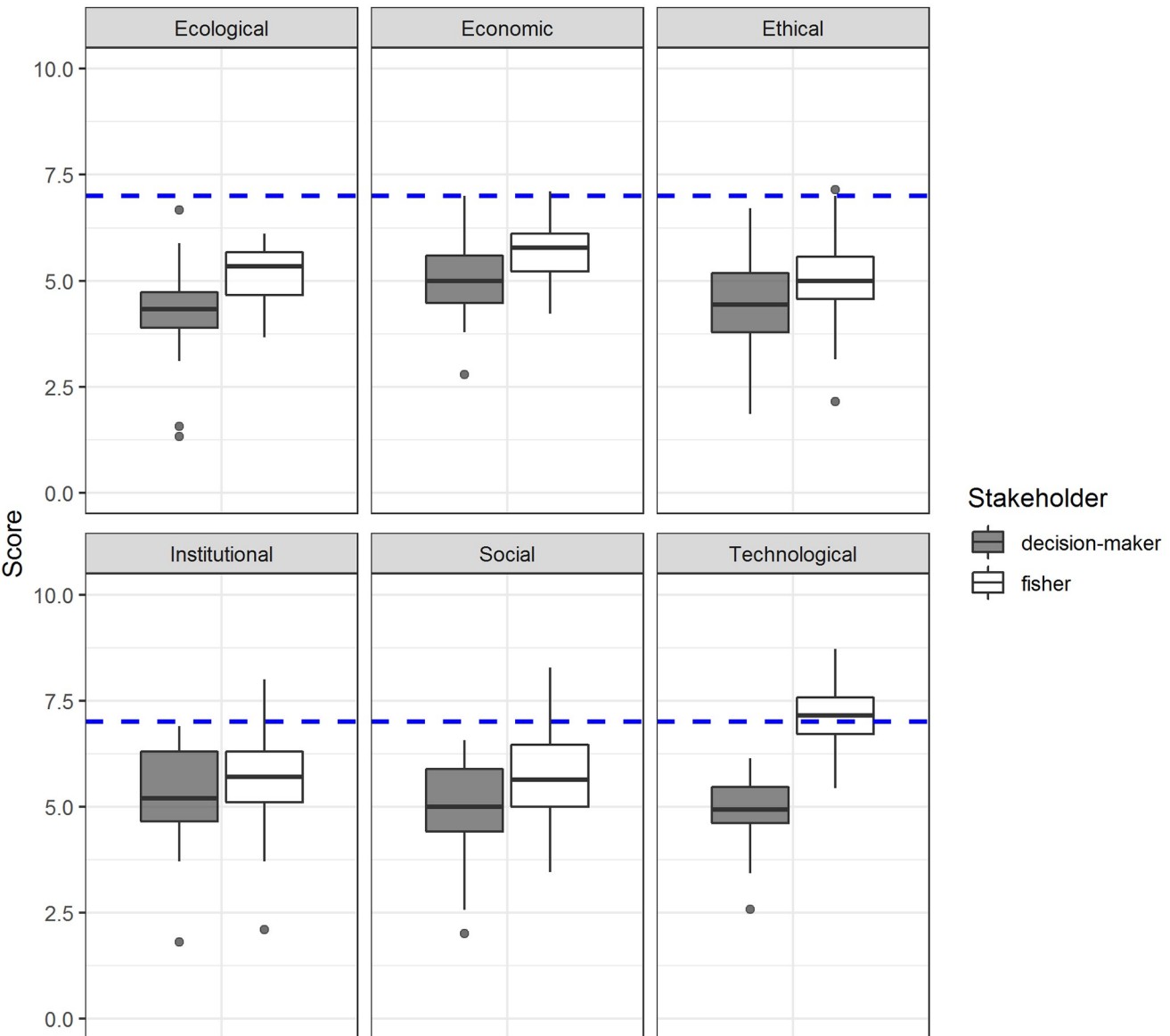

**Fig 2. Comparisons of the mean scores between fishers and decision-makers perceptions for the six dimensions.** Scoring scale is from zero (worst) to 10 (best), a score of 7/10 indicates a good score (blue dashed line), based on Rapfish (Pitcher et al., 2013).

### The influence of individual attributes on the transdisciplinary dimensions

In the previous sections, we outlined the different perceptions of the importance of different Rapfish dimensions according to different (human, physical, infrastructure) variables. Each Rapfish dimension consists of several attributes on which the dimension was scored. For instance, the average score of the social dimension was made up of individually scored attributes such as the strength of social networks, leadership replacement, or fishers' participation in the organization (S3 Table).

We applied a leverage analysis to detect the influence of individual attributes on the average score for a dimension (Fig 3A–3F). The standard error range in values (S.E. %) is between 2% and 6% [122]. Our results indicate that the influence of most attributes is less than 6%. A

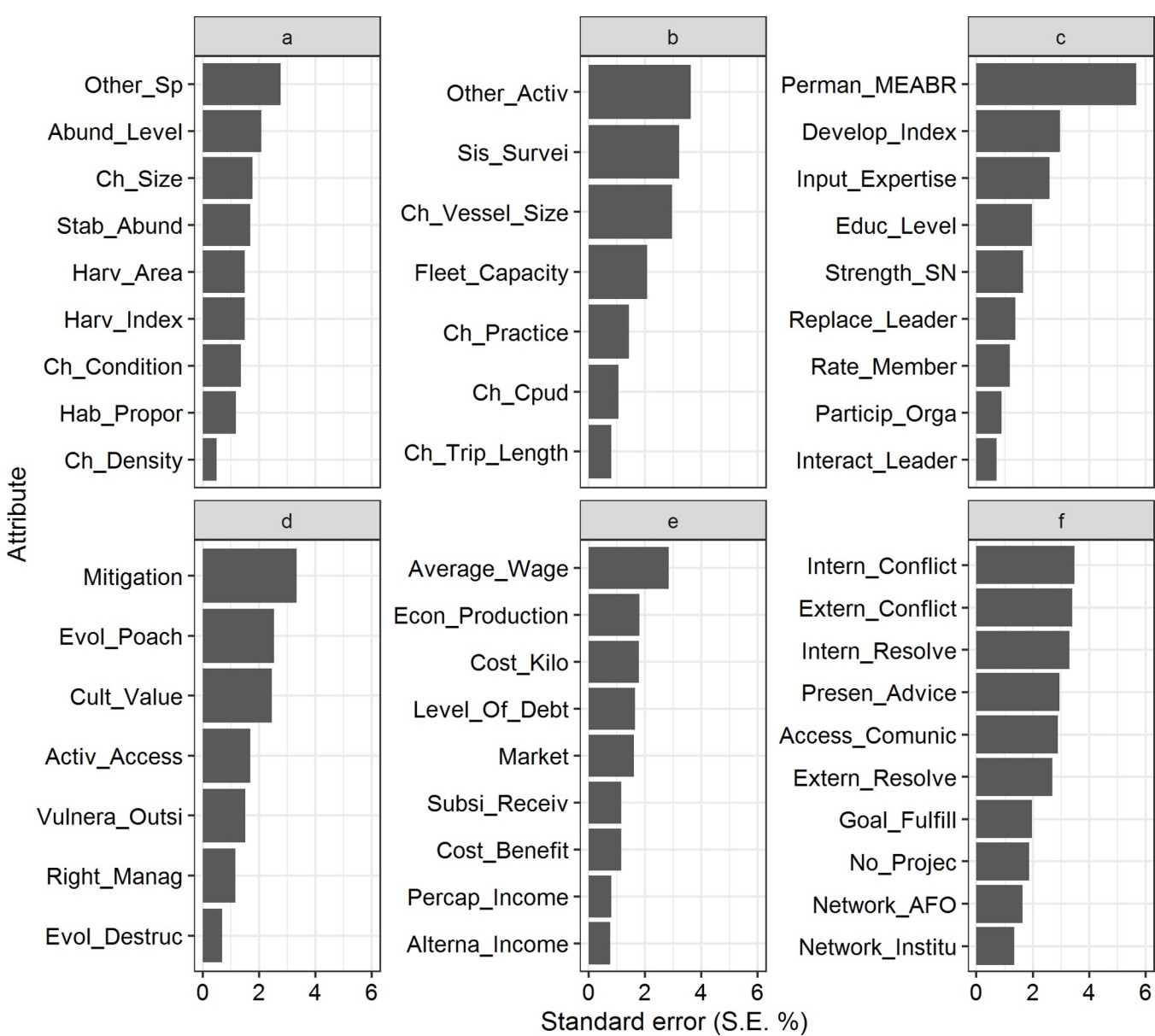

**Fig 3. Leverage of attributes, given by standard error (S.E. %), in each Rapfish dimensions for MEABR system.** (a) ecological, (b) technological, (c) social, (d) ethical, (e) economic and (f) institutional dimensions. Root mean square % change in ordination position when selected attribute removed (on status scale 0 to 100).

notable exception is the influence of the tenure of the management area ('Perman_MEABR') which is an attribute in the social dimension, with more permanency of tenure having a more favorable score (S.E. 5.66%). We observed the lowest values for the ecological dimension (Fig 3A) with leverage ranging from 0.49% to 2.77% (S.E.). Therefore, all attributes have approximately the same level of influence (same level of importance) on the average score for this dimension, except the re-orientation to other species ('Other_Sp'). For the technological dimension (Fig 3B), both the secondary effects from other extractive activities in the MEABR's ('Other_Activ' attribute) and system of surveillance ('Sis_Survei' attribute), had a more significant influence on the overall score (S.E 3.63% and 3.22%, respectively).

Regarding the ethical dimension (Fig 3D), the 'Mitigation' attribute was the most influential (S.E. 3.33%), and values for other attributes decreased gradually. As far as mitigation was concerned, they acknowledge that not much is being done (both by fishers' organizations and the government) in terms of recovery plans for damage caused to the MEABRs, whether by natural or man-made causes.

The average wage ('Average_Wage') had the most significant influence on the economic dimension (S.E. 2.85%) (Fig 3E). However, this wage represents the main income for some fishers and a complementary income for other fishers depending on the harvested species. For example, fisherwomen collecting seaweeds dedicate themselves exclusively to this activity. On the other hand, fishers extracting Chilean abalone consider the MEABR system as an 'extra activity' since most of them get highly variable payments ranging from 100,000–250,000 Chilean pesos (CLP) (around 300 US\$/fishing season see S2 Text, for more detail). Conflict–both internal ('Intern_Conflict') and external ('Extern_Conflict')—and internal conflict resolution ('Intern_Resolve') had the most influence in the institutional dimension (Fig 3F). All these attributes are indicators of the quality of relationships between users. These users face conflict due to different potential deficiencies of the governance to the system. The heterogeneity of fishers' beliefs and external aspects such as limitations to access, poaching, and illegal fishing guided these differences.

There were some notable demographic differences in the scoring of individual attributes within each dimension, concerning gender, organizational role, and MEABR accessibility (see S4 Table, S1–S3 Figs for details). Females assigned significantly higher levels of importance to re-orientation to other species ('Other_Sp') (S.E. 4.10%) and the average wage ('Average_Wage') (S.E. 4.00%) in the outcome of the ecological and economic dimension respectively. In Chile, participation by women acquired greater importance (and thus higher potential wages) in MEABRs due to seaweed harvesting activities, despite artisanal fishing being considered a strictly male activity for a long time. In other words, the involvement of women has enhanced their increase and interest, has allowed more significant personal development and higher economic contributions to their homes.

Differences in fishers' responsibility influence their ability to decide on changes and internal management systems in fishers' organization (including the cohesiveness of the organization). In terms of technology, the secondary effect of other activities ('Other_Activ') (S.E. 5.91%) and the surveillance system ('Sis_Survei') (S.E. 4.23%) were more important to leaders' perceptions of technological outcomes.

The accessibility of the MEABRs (measured as distance) can be related to the threat of poaching, i.e. in more distant MEABRs it is more difficult to enforce control, according to several authors [68, 77, 123]. For the more accessible MEABRs only four attributes were more sensitive among the six dimensions described (i.e. 'Other_Sp', 'Abund_Level', 'Mitigation' and 'Sis_Survei'). Regarding the technological dimension, the presence of surveillance ('Sis_Survei') was considered most influential in the more accessible MEABRs (<0.5km).

**Communication level between member fishers and other 'actors'.** Member fishers (n = 86) acknowledge having relations with other groups through cooperative contacts and even conflicts. All members were asked: How is your level of communication with leaders and other actors (artisanal fishers' organizations and decision-makers)? Member fishers have different relationships with other groups, such as positive collaborative relationships or more negative relationships around conflict. Member fishers had the most positive and intense relationship with leaders (7.22 ± 2.3, Fig 4). There seem to be two distinct groups regarding the intensity and quality of the relationship between member fishers and leaders, because 26.7% (0–3 score) of member fishers indicated that their relationship with leaders was not intense and not good (lacking communication and working relationships). The relationships with

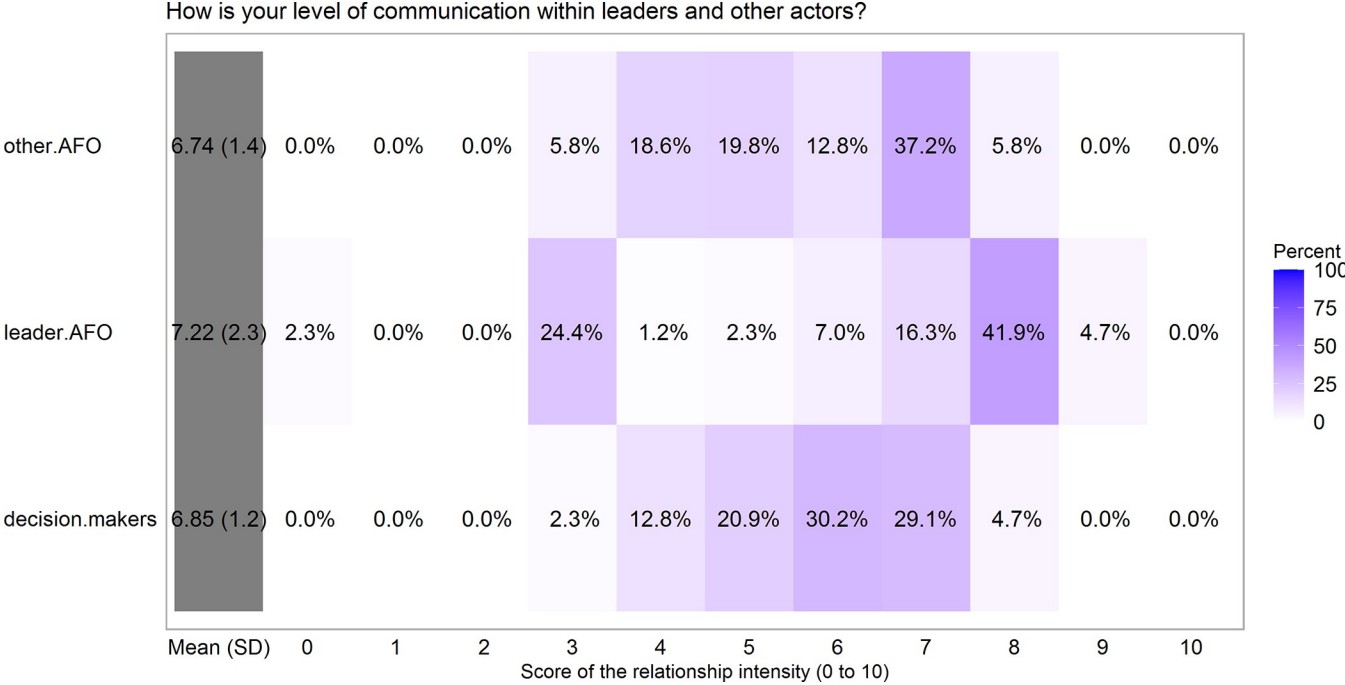

**Fig 4. Percentage and mean scores (±*SD*) of the level of communication between member and other actors.** Other actors are leaders of their Artisanal Fisher Organizations (AFOs), non-fishers, and other AFOs of the MEABRs in Biobio Region. Scoring scale is from zero (worst) to 10 (best), a score of 7/10 is a good score.

other fisher organizations and decision-makers ranged from moderate to good (6.74±1.4 and 6.85±1.3; respectively).

## Discussion

In general, the MEABR system has continually evolved and adapted to a new socio-ecological reality [81]. Adaptation is vital if the MEABR system is to be successful in the long term [42]. Including socio-ecological knowledge in the management process would help in resource co-management, improving the environmental aspects, and the human component (i.e., fisher organizations, scientists, managers, and government institutions). Despite the effort spent on establishing MEABRs and management policies, fishers' perception and behavior towards the MEABRs have not received enough attention. In this context, our study focused on the perception of artisanal fishers and their knowledge of MEABR performance.

### Gender perspective

An interesting perspective of our findings was that fisherwomen and fishermen had different views. On the one hand, the fishermen highlighted the importance of the ecological and economic dimensions of MEABR's. On the other hand, fisherwomen stated that the institutional dimension was the most significant. For example, for the fisherwomen in Coliumo Bay (Biobio Region) artisanal fishing has improved gender roles and influenced decisions about coastal resources access and use [67, 88]. The views of women have become more important in fishery management, for example, where women have increased their capacity, confidence, and engagement for good fishery practices [124].

Two of the seaweed gatherers fishing unions that were surveyed were comprised of women (located in Coliumo Bay). A fisherwoman (leader) stated that they do not compete with men but support them:

"Women have a much broader view. . . [in the way they] look for ways to generate resources. . . but that has not always been perceived positively by men. If men perceive women with these characteristics and this type of profile more positively, women should be accepted as co-workers more easily. But [the fishermen] do not perceive women as contributors, but as competitors. We are not in competition. We want to be supporters and partners, not to stay behind (fisherwoman leader #30)".

A number of the fishermen remarked on women's participation as union members only:

"The women are only members of the organization, they do not join in the harvest process, they only support with monetary contributions such as 'fees' (fisherman leader #7)".

However, some of them mentioned that women's management skills when assuming a leadership positions was favored:

"In a short time, she has gotten two projects that never happened before. . . women have another way of thinking (fisherman member #9)".

From the above, we highlight two visions of women's participation, not only as members of an organization but also when they assume leadership. For this reason, the responsibility of a MEABR under female administration has addressed a challenge, not only by involving them in this activity but also by empowering them in new roles. Women's perspectives have been very influential in an institutional context, and it is reflected in the permanence of the MEABR. However, women's participation in fishing unions is still low, with only 10% holding positions as president, secretary, or treasurer [102].

MEABR's that are run by women also have an ecological focus with a reorientation to seaweed collection. This could be due to the depletion of the main target species, and most of the women surveyed are from those MEABR where a reorientation has occurred. Seaweed collection is an activity generally led by women, but it is important to reiterate that it is also a family activity.

Most women (like men) consider the MEABR, a complementary activity, and a high percentage of them are household heads or homemakers who undertake other social support activities, such as "Pro-employment" [91]. Few studies have focused on women at the local level in Chile, but gender equality approaches in artisanal fisheries are available from other parts of the world. Women's actions can lead to more sustainable fishing, and their organizational capacity can empower them, not only as fisherwomen and workers but also as resource managers [86, 88, 125, 126].

## Organizational roles perspective

Fisher leaders and members had different perspectives on the social outcomes of MEABRs with regard to the permanence of management (i.e. continuity and efficiency of the MEABR's performance). Although both abundance and harvests have decreased in recent years in most management areas of Biobio Region (ITA reports), members and leaders have recognized that MEABRs provide them with additional benefits. For example, during periods of low fishing activities in other fisheries in which they participate, i.e., fishing for sardine, anchovy, hake, or

jumbo squid, they can still harvest the benthic target species. The MEABRs do not seem to be an economic solution for fishers; instead, they complement fishers' livelihoods [68].

> "In bad times, it helps us. . . we give work to guys who want to join the organization and want to get their 'luquitas en la Pega' ('money with work'). . . when there is a shortage of people. We invite them, and they are involved in the work. The resource is essential for us, also for the 'Caleta' (fishing cove), so the 'Caleta' is becoming quite well known (fisherman leader #14)".

> "It is an income in wintertime, which is the 'bad period' in the fishery (fisherman member #16). We have months when we get products only from the management areas, so we learn to take care of the resources (fisherman member #35)".

Despite this positive perception, the MEABR system does not fully meet the fishers' expectations [54]. Fishers often express concerns about low incomes. Their low profits are also affected by poaching (whether by outsiders, non-unionized fishers, or even by members). Poaching is one of the main factors influencing the scarcity of the target resource and one of the most critical problem perceived by fishers [77, 79, 81, 106, 127, 128]. Thus, ignoring illegal activity outcomes in false conclusions (over- and under-representation) about the state and trends of fisheries; it also adds to the increased costs associated with surveillance and reduced income for fishers [78]. Consequently, this leads to conflicts among members, with some leaving the association altogether.

> "Now it is not profitable, only in the first 'sacas' (harvests). Before, we earned 500,000 Chilean pesos. After that, it is barely 200,000 or less because the 'loco' has diminished. All of this is ending because of overexploitation (fisherman member #63)".

On the one hand, the members surveyed had a negative perception of the continuity of the MEABR due to the association's lack of cooperation and commitment, decreasing the level of trust either among members towards leaders or towards other stakeholders:

> "It is challenging if there is no union, the management area can collapse because of the deficiencies observed (fisherman member #9)".

On the other hand, most of the leaders surveyed hope that the MEABR will improve and overcome the deficiencies generated since they were assigned:

> "I want to go back when the management area was first introduced on the island. I know that it will be difficult, very hard. It depends on the authorities, on the management of the leaders themselves, and the fishermen (fisherman leader #15)".

In order to promote the continuation of artisanal fishing within the coastal management plans, interaction among all stakeholders needs to be improved [129]. Despite the lack of communication, it is necessary to reach an agreement gradually. There are also proposals related to "collective" commercialization among fisher organizations [54]. Nowadays, few associations are working together to turn the management areas into either enterprises or cooperatives because that requires commitment from all sides.

## The accessibility of the MEABRs

Our survey asked the following question: "are more distant areas harder to manage?" We found that the perceptions of respondents about accessibility were significant (see S4 Table). The ecological outcomes were better for less accessible areas (>0.5 km), mainly those located on islands. Our results can complement the study conducted by Andreu-Cazenave et al. [4], showing that the MEABRs located farther from the fishing coves 'caletas' are more naturally protected by environmental conditions (i.e. most exposed zones) in addition to a lack of access roads. These factors allowed the most valuable resource to register lower mortality, and hence less susceptibility to poaching. In contrast, those MEABRs with easy access were more vulnerable to poaching, and consequently registered the highest target species mortality. We can complement this topic from the findings of Cinner et al. [117] and Maire et al. [130], who showed that reef fish biomass is strongly related to the accessibility of both markets and local communities. This means that market proximity affects fishing gear (technique effect), wealth, and selling strategies (scale effect) of coastal communities. In our case, we considered that other drivers could influence the performance of the MEABRs due to their variability and the effect of open access areas in the region, which are under-researched in our study zone.

However, the problem of illegal fishing (poaching) is common in MEABRs, whether distant or easily accessible. The impact of illegal fishing has increased conflicts in many MEABRs [42, 77, 131] and even some fisher organizations authorize their members to fish illegally within their management areas [106].

In general, illegal fishing is a problem that continues to increase worldwide and it is difficult to estimate accurately [132]. It is distributed in a complex network of relationships, practices, and actors embedded in a geographical and cultural context [133]. And the poaching risk has been better predicted when drivers related to fishers' spatial preference (i.e accessibility and attractiveness) interacted with fishing capacity [119]. In the case of slow-moving sedentary resources, like the majority of species included in the management and exploitation plans, their stock assessments become a challenge due to the combined effect of illegal fishing and their irregular distribution, as in the case of South African abalone [134]. Due to the natural aggregation patterns of these resources at different spatial scales, the accuracy of divers' estimations is reduced, and their overexploitation is accelerated [135].

## Conclusion

This study shows that the perceptions of different stakeholders about the MEABR system influence their local fisheries management expectations in Biobio Region. We observed that individuals with differing gender and organizational role, and the accessibility of the MEABR influenced the expectations of this current system.

More than half the fishers were dissatisfied with the MEABR system in terms of illegal fishing practices (poaching), low productivity, low profits, and conflicts inside and outside the fishers' organizations. They agreed that the enforcement of punitive measures is an effective means to control illegal fishing practices. Fishers emphasized inadequate monitoring and control systems, and they stressed the lack of commitment to the management areas and the lack of conservation species.

In terms of conflicts, most fishers admit that strengthening social networks within their groups improves communication. Fishers recognize that the most common cause of friction generated in this system is poaching. This problem is very hard to solve in the short-term and, generally, conflicts have been increasing in many MEABRs along the Chilean coast.

The ecological and social dimensions were most influential in shaping the fishers' perceptions of successful MEABRs. Based on this, there is a dynamic exchange between these

approaches and their complexity in the system. Our results showed limited communication and relationships between fishers and other stakeholders, which makes fisher's organizations weak and dependent on external sources. Therefore, government managers and decision-makers should enhance and increase relational capacities (within organizations, among organizations and between organizations and the local environment) to improve their social capital on a more permanent basis as viable alternatives to their current modes of fishing. Besides, in the future, the government must increasingly include the fishers' local knowledge and management strategies to ensure the best local understanding. Only a strong fisher-government partnership would help solve weaknesses in the system, promote continuity and facilitate responsible fisheries in MEABRs.

Beyond the local knowledge and fishers' perceptions, our study allowed us to understand a little more about how the MEABR system has developed in the region. We acknowledge that most fisheries' problems are complex and contain human and ecological dimensions, and as such, we deal with it in a fragmented way. We argue that our insights provide ideas on how to formulate appropriate strategies for the management of the MEABR system, especially in forging improved communication and strengthening the social networks among the whole stakeholders' community, not only at the local level but also between the different regions of Chile.

## Supporting information

**S1 Fig. Leverage analysis for attributes of six Rapfish dimensions (standard error S.E. %) by gender.**
(TIF)

**S2 Fig. Leverage analysis for attributes of six Rapfish dimensions (standard error S.E. %) by fishers' organizational roles.**
(TIF)

**S3 Fig. Leverage analysis for attributes of six Rapfish dimensions (standard error S.E. %) by MEABR accessibility (Accessible and Less accessible).**
(TIF)

**S1 Table. Description of the four zones in the Biobio Region identified in this study.**
(PDF)

**S2 Table. Characteristics of each Management and Exploitation Area of Benthic Resources (MEABR) selected in the Biobio Region.**
(PDF)

**S3 Table. Rapfish sustainability dimensions and attributes adapted for the MEABR.**
(PDF)

**S4 Table. Average fishers' perception scoring for all 18 variables (see note below table for description) per each Rapfish dimension.** One-way ANOVA analysis and t-test comparison between respondents (fishers) categorized.
(PDF)

**S1 Appendix. Questionnaire for fishers (leaders).**
(PDF)

**S2 Appendix. Questionnaire for fishers (members).**
(PDF)

**S3 Appendix. Questionnaire for non-fishers (decision-makers, academics and consultants).**
(PDF)

**S1 Text. Selection, stratification and sampling design.**
(DOCX)

**S2 Text. Some details about payments.**
(DOCX)

## Acknowledgments

We sincerely thank leaders and members of the fisher organizations from the MEABRs in Bio-bio Region for their dedicated time, trust, and collaboration. We express our gratitude for the participation of professionals and students, who supported us during the fieldwork. Likewise, we want to thank professionals from SUBPESCA, SERNAPESCA, IFOP, academy, and private consultants for providing valuable information during the interviews.

Thanks to CSIRO Oceans and Atmosphere for hosting MFM under the supervision of Dr. Ingrid van Putten and make her internship an excellent opportunity for learning and professional development.

## Author Contributions

**Conceptualization:** Milagros Franco-Meléndez.

**Data curation:** Milagros Franco-Meléndez.

**Formal analysis:** Milagros Franco-Meléndez.

**Funding acquisition:** Luis A. Cubillos.

**Investigation:** Milagros Franco-Meléndez.

**Methodology:** Milagros Franco-Meléndez.

**Project administration:** Luis A. Cubillos.

**Supervision:** Jorge Tam, Ingrid van Putten, Luis A. Cubillos.

**Writing – original draft:** Milagros Franco-Meléndez.

**Writing – review & editing:** Milagros Franco-Meléndez, Jorge Tam, Ingrid van Putten, Luis A. Cubillos.

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
