## [Decision Letter · Decision Letter 0]

22 Dec 2020

PONE-D-20-30809

Integrating human and ecological dimensions: Importance of fishers' perceptions and participation on the MEABR performance

PLOS ONE

Dear Dr. Franco Meléndez,

Thank you for submitting your manuscript to PLOS ONE. After careful consideration, we feel that it has merit but does not fully meet PLOS ONE’s publication criteria as it currently stands. Therefore, we invite you to submit a revised version of the manuscript that addresses the points raised during the review process.

I tend to agree with the comments and suggestions made by reviewer #1, the manuscript needs to be re-casted following the steps 1-3 described by reviewer #1, and in the introduction you need to clearly explain the needs and provide evidence for such study as well as the theoretical background with the respective cited references. 

We look forward to receiving your revised manuscript.

Kind regards,

Andrea Belgrano, Ph.D.

Academic Editor

PLOS ONE

Journal Requirements:

3.We note that [Figure(s) 1] in your submission contain map images which may be copyrighted. All PLOS content is published under the Creative Commons Attribution License (CC BY 4.0), which means that the manuscript, images, and Supporting Information files will be freely available online, and any third party is permitted to access, download, copy, distribute, and use these materials in any way, even commercially, with proper attribution. For these reasons, we cannot publish previously copyrighted maps or satellite images created using proprietary data, such as Google software (Google Maps, Street View, and Earth). For more information, see our copyright guidelines: http://journals.plos.org/plosone/s/licenses-and-copyright.

1.    You may seek permission from the original copyright holder of Figure(s) [1] to publish the content specifically under the CC BY 4.0 license. 

4. Please remove your figures from within your manuscript file, leaving only the individual TIFF/EPS image files, uploaded separately.  These will be automatically included in the reviewers’ PDF.

Reviewers' comments:

Reviewer's Responses to Questions

**Comments to the Author**

1. Is the manuscript technically sound, and do the data support the conclusions?

Reviewer #1: Partly

2. Has the statistical analysis been performed appropriately and rigorously? 

Reviewer #1: Yes

3. Have the authors made all data underlying the findings in their manuscript fully available?

Reviewer #1: Yes

4. Is the manuscript presented in an intelligible fashion and written in standard English?

Reviewer #1: No

5. Review Comments to the Author

Reviewer #1: This manuscript provides an interesting study of a necessary issue: how to incorporate the social dimension of marine resource management. The authors provide a well-rounded case study in Chile. However, the study has some important methodological shortcoming. Moreover, is poorly written which makes it very challenging to judge its merit. I would recommend authors to seek editorial assistance before re-submitting this article. I provide some overall comments and then some specific comments.

1) There is an absence of a theoretical background in the introduction (this is clearly reflected in the abstract which does not contain a single line referring to why there is a need for such a study). Understanding the perception of those participating in fisheries is clearly important and this is vaguely stated in the first two paragraphs. But there has been extensive work looking at how perceptions affect performance in fisheries settings, which the authors do not reference. I would encourage authors to re-write the introduction including theoretical perspectives on what they are measuring (e.g. role of gender). Otherwise, it becomes quite challenging for the reader to understand why you did the analysis you did

2) The methodological approach has some important shortcoming that need to be reconsidered. The grouping strategy is very unclear, not only in how authors did it, but also on why. Moreover, the sample size calculation is misleading as they only consider the overall population of fishers, when their design is really stratified into MEARBs. I provide more details on this below

3) Overall, the manuscript is very poorly written. There are sections that don’t seem to have been edited at all (e.g. lines 67-78: “Several fisheries worldwide are now using TURFs as a management strategy in several fisheries around the world”) while others are very unclear, and the English needs to be polished. I provide several examples in the specific comments.

Specific comments:

Introduction

- Paragraph 38-47: I think this paragraph could be re-written to more clearly make the point you are trying to make. Are you trying to say that there is a threat of overfishing that needs to be considered? In which fleets? Does the focus need to be put in the social realm of fisheries management?

- Line 51-53, which impacts?

- Line 51, it is not clear what the authors mean by “weaknesses in our knowledge of human responses”

- Line 53-55 Provide references

- Line 53-55 authors already state something similar in the first paragraph. Both these paragraphs need a deep revision, as to make clear what is the message. As written it is very difficult to understand the introduction.

- Line 60: “Avoid marine resource issue” please re-write

- Lines 60-66: The text could be improved substantially if authors better explain this system. A few notes: what do you mean by “set the rules …..for fishers performance”?, in ii) authors sort of contradict themselves with i)….who sets the guidelines? (and what do they mean by guidelines?), government is responsible for enforcement, not compliance, iii) is unclear (how can only fishers execute co-management, which involved the government?)

- Lines 72-73: are those studies focusing on social aspects or the AMERB themselves. Very unclear

- Lines 76-77: Durvillaea is not bull kelp

- Line 79-81: References?

- Line 82: Can you expand on this?

- Line 84: you mean collective action?

- Line 95-97: By the time the reader gets to this, there has been absolutely no background on the theoretical perspectives for hypothesising this. The paper presents background on the system itself, but not on previous work that links individual characteristics and behaviour. This is a very hot research topic and there are dozens of studies that deal with this. I would strongly encourage authors to do a deeper literature review and re-frame their hypothesis/research questions based on what others have done in the past. As written, the introduction only states that the human dimension of management needs to be considered. This is OK, but only as a starting point, it does not suffice as a theoretical background.

Methods

- Line 116-118: Phrase not finished

- Line 130: should be differences

- Line 130-134: Please explain this a bit more clearly and provide references. What do you mean by “accessibility had implications for its performance”

- Lines 135: Please explain what do you mean by decreed

- Table 1, please fix last column

- Line 149: Why were those included and not others?

- Line 152: does that mean those are closed now?

- Authors don’t explain why they have done this grouping and why they arbitrarily decided to replace some AMERB

- Moreover, it is very unclear HOW this grouping was developed. Very unclear section

- Line 173, why do you consider a different number than what’s stated in Table 2? (1080 vs 1230)

- Authors seem to ignore the fact that this this is a stratified sample, is not just one population, is a series of groups that you are interested in. As such, just aggregating and obtaining one sample size is not the appropriate way to do it if you want to say something about the differences between groups. Moreover, they seem to ignore the fact that there are different population sizes in each group.

- Line 190-191: Please describe in more detail and better justify the inclusion of decision-makers

- Line 192-193: But how was these data analysed?

- Line 196-197: Why include the central zone?

- Line 197: Please re-write

- Several of the attributes assessed and presented in table S3 are group level attributes. Why did you assessed these at the individual level, while the response is a group level variable (e.g. abundance of target species, fleet size)

- Lines 225-233: it is still unclear why you did this

- Line s234-238: Authors do no explain why they chose to divide fishers into these groups and why is it interesting to assess these (e.g. why would you expect differences between those above and below 50 years old?). The way authors go about their methodology seems arbitrary and not theoretically informed, which makes it very challenging to follow what they are trying to do.

Results

- Line 181: Re-write subtitle as it’s not very clear

- Lines 282-286: this should go in methods

- Lines 362-363: Can you provide references for this? Authors state this as obvious, but it is not entirely clear to me if this is so (and I think by manage they mean enforce)

- Line 363-364: phrase is incomplete

Discussion

- Line 384-385: It is not entirely clear where this statement is coming from

- Line 386-388: This is a key statement, but should go in the introduction and better explained and referenced

- Gender perspective. This is a very interesting section, but requires a better theoretical grounding in the introduction so that readers understand from the beginning that assessing this was an objective of the study

- Position perspective. It is not entirely clear what do authors refer to by “position perspective”. Is it the role MEARBs play in their livelihood?

- Line 459: please provide references, there are several studies looking at this, specifically for MEARBs in Chile. It would be helpful if you could expand on this, as it is a critical management issue

- Line 475: This line starts with a reference, which is unclear

- Line 485-489: Authors could also look at the work by Cinner and others (e.g. Disentangling the complex roles of markets on coral reefs in northwest Madagascar) to complement this (distance to market). But it is also important to consider that those MEARB that are more accessible are easier to enforce, so there is no simple explanation for this distance/protection relationship

Conclusion

- Lines 508-512: I am not entirely sure the data supports this hypothesis. Certainly the first section, but not the willingness to use the resource more sustainably

- Line 526-228: Very unclear. Why are these organisations characterised as fragile and dependent? What coaching do you refer to?

- Line 528-530: I am not entirely sure how this comes out of the results, please clarify

Examples of lines that need re-writing

42-43

67-68

73-74

191-192

6. PLOS authors have the option to publish the peer review history of their article (what does this mean?). If published, this will include your full peer review and any attached files.

Reviewer #1: **Yes: **Rodrigo Oyanedel

---

## [Author Response · Author response to Decision Letter 0]

4 May 2021

We would like to thank the Editor-in-Chief and the reviewer #1 for providing critical comments of our research. We have given careful consideration to each comment, resulting in a thorough revision of our text and several key modifications.

---

## [Decision Letter · Decision Letter 1]

3 Jun 2021

PONE-D-20-30809R1

Integrating human and ecological dimensions: The importance of stakeholders’ perceptions and participation on the performance of fisheries co-management in Chile

PLOS ONE

Dear Dr. Franco Meléndez,

Thank you for submitting your manuscript to PLOS ONE. After careful consideration, we feel that it has merit but does not fully meet PLOS ONE’s publication criteria as it currently stands. Therefore, we invite you to submit a revised version of the manuscript that addresses the points raised during the review process.

In your revision please address all the remaining comments suggested by reviewer #1, that also in my opinion will add clarity to the manuscript.

We look forward to receiving your revised manuscript.

Kind regards,

Andrea Belgrano, Ph.D.

Academic Editor

PLOS ONE

Journal Requirements:

Reviewers' comments:

Reviewer's Responses to Questions

**Comments to the Author**

1. If the authors have adequately addressed your comments raised in a previous round of review and you feel that this manuscript is now acceptable for publication, you may indicate that here to bypass the “Comments to the Author” section, enter your conflict of interest statement in the “Confidential to Editor” section, and submit your "Accept" recommendation.

Reviewer #1: All comments have been addressed

2. Is the manuscript technically sound, and do the data support the conclusions?

Reviewer #1: Yes

3. Has the statistical analysis been performed appropriately and rigorously? 

Reviewer #1: Yes

4. Have the authors made all data underlying the findings in their manuscript fully available?

Reviewer #1: (No Response)

5. Is the manuscript presented in an intelligible fashion and written in standard English?

Reviewer #1: Yes

6. Review Comments to the Author

Reviewer #1: I congratule authors for this new version of the manuscript. It has improved considerably. However, I still think there are some issues that need a bit of work. Nothing substancial, but small things that would certainly improve the quality of your work, specially in the intro. Please see below:

-Paragraph 68-78 could be improved for clarity so that the original objectives of the AMERB system, and its current challenges are better differentiated. I would also recommend to comment on this work: https://www.ingentaconnect.com/content/umrsmas/bullmar/2017/00000093/00000001/art00006 and https://conbio.onlinelibrary.wiley.com/doi/full/10.1111/conl.12637

Also, on this paragraph, the "this is the main problem" is not clear. Which one is the main problem? This paragraph still needs a bit of work, because is pivotal in your study, so needs to be crystal clear.

-Paragraph 79-92 needs some editing. First, the MEARB system was conceptualized and some pilots started in 1990. It didn't gain traction until later. I also think you could be a more nuanced about the benefits/limitations and the overall performance of the system (see references above), as there are lots of successful cases. Now it reads as if the benefits are the exception.

-Paragraph 93-103 I think you mean "have not been met" in the second line. I would also say outcomes instead of results in the manuscript. The last sentences is a bit confusing (needs re-writing) as you jump quite unexpectedly from the local to worldwide

Paragraph 115-122 the reference provided above does exactly that systematic documentation, so would be good if you could comment on that

- I think that there still isn't a proper section of the intro that talks about gender. There's a lot of literature on this which the authors could look at, which consider the different types or roles women have played, how those roles have been neglected and pathways to their inclusion. This becomes very relevant in the face of the newly approved law in Chile on gender in the fisheries sector, so I would strongly recommend authors to go a bit deeper here

-Lines 182-186 please explain what an ESBA proposal is

7. PLOS authors have the option to publish the peer review history of their article (what does this mean?). If published, this will include your full peer review and any attached files.

Reviewer #1: **Yes: **Rodrigo Oyanedel

---

## [Author Response · Author response to Decision Letter 1]

28 Jun 2021

We would like to thank the Editor-in-Chief and the reviewer #1 for providing critical comments of our research. We provide clarification on the introduction section in the revised manuscript.

---

## [Editor Report · Decision Letter 2]

1 Jul 2021

Integrating human and ecological dimensions: The importance of stakeholders’ perceptions and participation on the performance of fisheries co-management in Chile

PONE-D-20-30809R2

Dear Dr. Franco Meléndez,

We’re pleased to inform you that your manuscript has been judged scientifically suitable for publication and will be formally accepted for publication once it meets all outstanding technical requirements.

Kind regards,

Andrea Belgrano, Ph.D.

Academic Editor

PLOS ONE

---

## [Editor Report · Acceptance letter]

2 Aug 2021

PONE-D-20-30809R2 

Integrating human and ecological dimensions: The importance of stakeholders’ perceptions and participation on the performance of fisheries co-management in Chile 

Dear Dr. Franco-Meléndez:

I'm pleased to inform you that your manuscript has been deemed suitable for publication in PLOS ONE. Congratulations! Your manuscript is now with our production department. 

Kind regards, 

on behalf of

Dr. Andrea Belgrano 

Academic Editor

PLOS ONE